

# Poor sleep quality association with higher lung cancer risk: a nested case-control study

Guo-Tian Ruan[1,2,3,*], Ya-Ping Wei[4,*], Yi-Zhong Ge[1,2,3], Li-Shun Liu[5,6], Zi-Yi Zhou[5,6], Sultan Mehmood Siddiqi[5], Qiang-Qiang He[5], Shu-Qun Li[6], Jia-Feng Xu[6], Yun Song[7,8], Qi Zhang[1,2,3], Xi Zhang[1,2,3], Ming Yang[1,2,3], Ping Chen[9,10], Yong Sun[11], Xiao-Bin Wang[12], Bin-Yan Wang[6,8] and Han-Ping Shi[1,2,3]

[1] National Clinical Research Center for Geriatric Diseases, Xuanwu Hospital, Capital Medical University, Beijing, China
[2] Key Laboratory of Cancer FSMP for State Market Regulation, Beijing, China
[3] Department of Gastrointestinal Surgery/Department of Clinical Nutrition, Beijing Shijitan Hospital, Capital Medical University, Beijing, China
[4] College of Public Health, Shanghai University of Medicine and Health Sciences, Shanghai, China
[5] Graduate School at Shenzhen, Tsinghua University, Shenzhen, China
[6] Shenzhen Evergreen Medical Institute, Shenzhen, China
[7] AUSA Research Institute, Shenzhen AUSA Pharmed Co Ltd, Shenzhen, China
[8] Institute for Biomedicine, Anhui Medical University, Hefei, China
[9] Inspection and Testing Center, Key Laboratory of Cancer FSMP for State Market Regulation, Shenzhen, China
[10] College of Pharmacy, Jinan University, Guangzhou, China
[11] The First People's Hospital of Lianyungang City, the First Affiliated Hospital of Kangda College of Nanjing Medical University, Lianyungang, China
[12] Department of Population, Family and Reproductive Health, Johns Hopkins University Bloomberg School of Public Health, Baltimore, USA
* These authors contributed equally to this work.

Corresponding authors
Han-Ping Shi, shihp@ccmu.edu.cn
Xiao-Bin Wang, xwang82@jhu.edu

## ABSTRACT

**Background:** Little is known about the relationship between sleep quality and lung cancer incidence. Thus, this study was conducted to investigate the potential connection between sleep quality and lung cancer incidence.

**Methods:** We performed and selected a nested case–control study that included 150 lung cancer cases and 150 matched controls based on the Lianyungang cohort. Univariate and multivariate logistic regression was utilized to investigate the connection between potential risk factors and lung cancer incidence risk.

**Results:** In this study, the average age of participants was 66.5 ± 9.1 years, with 58.7% being male, and 52.7% reportedly experiencing sleep quality problems. The results of multivariate logistic regression showed that poor sleep quality was connected to an increased lung cancer incidence risk ($P = 0.033$, odds ratio = 1.83, 95% confidence interval = [1.05–3.19]) compared with those with good sleep quality. The stratified analyses showed a significantly positive connection between poor sleep quality (*vs.* good sleep quality) and cancer risk in smokers (*vs.* non-smoker, *P* for interaction = 0.085). The combined effect analysis indicated that smokers with poor sleep quality suffered from a 2.79-fold increase in cancer incidence rates when compared with non-smokers with good sleep quality.

**Conclusions:** Poor sleep quality was positively connected to an increased lung cancer incidence risk. In addition, among those individuals with poor sleep quality, smoking increased the lung cancer incidence risk.

# INTRODUCTION

At present, non-communicable diseases are the primary cause of death worldwide. Cancer is expected to develop into the main death cause in the entire world in the 21st century, and one of the most significant reasons for lower life expectancy is cancer. According to the 2020 global cancer data, lung cancer ranks second among all cancer types in terms of new cases and ranks first in new deaths (*Sung et al., 2021*). Smoking is an important, significant lung cancer risk factor (*Humans IWGotEoCRt, 2004*; *Sasco, Secretan & Straif, 2004*). However, the lung cancer incidence risk among non-smokers in developing countries is rising (*Youlden, Cramb & Baade, 2008*). Other important risk factors (such as age, environment, occupation, and heredity) are contributing to the increased incidence of lung cancer. Even in China, the risk factors for lung cancer vary in different regions due to differences in environment and lifestyle (*Zhou et al., 2014*). Thus, there is a sore need to find out other potential and changeable risk factors to stop and reverse the rising incidence risk trend of lung cancer.

Sleep is a natural circulatory state of human beings, and it is a physiological phenomenon that can meet the basic physiological and psychological needs of human beings. Indeed, sleep plays an important part in human health and happiness, and affects physical development, physiological processes, emotional regulation, cognitive functioning, and quality of life (*Hirshkowitz et al., 2015*). Insomnia is one of the most common sleep abnormalities. Insomnia incidence in the United States rose from 17.5% in 2002 to 19.2% in 2012 (*Ford et al., 2015*). In recent years, in European countries, the proportion ranged from 6% to 19% (*Riemann et al., 2017*); in East Asia, 39.4% of adults in Hong Kong have poor sleep quality, compared to 13.1% in South Korea and 16% in Mainland China (*Haseli-Mashhadi et al., 2009*; *Wong & Fielding, 2011*; *Wu et al., 2018*). Abnormal sleep was reported to be connected to a variety of adverse health problems, including diabetes, obesity, cardiovascular disease, and even cancer (*Davis & Mirick, 2006*; *Markt et al., 2016*; *Wiggins et al., 2020*). Sleep quality has become a major problem facing human beings with the development of modern society.

Sleep abnormalities cause an obvious violation of the body's circadian rhythm (*Sack et al., 2007*). It has been reported that disruptions to circadian rhythms are connected to gene expression alteration, cancer biology, and cancer metabolism (*Savvidis & Koutsilieris, 2012*; *Takahashi, 2017*; *Verlande & Masri, 2019*). Poor sleep quality decreases floating melatonin in the system (*Soucise et al., 2017*). In previously reported mouse models of lung cancer, melatonin can reduce lung cancer cell proliferation and metastasis and induce cell apoptosis (*Ma et al., 2016*). It is well known that night shift work is related to poor sleep

quality, and it is connected to an increased incidence of several cancers, including lung cancer (*Parent et al., 2012*). To date, many previous studies have mainly focused on lung cancer survivors who suffered from poor sleep quality following their diagnosis of lung cancer (*Gottfried et al., 2020*; *Khawaja et al., 2014*). However, little is known about the connection between sleep quality and lung cancer incidence risk. Furthermore, it is reported that sleep quality is connected to cancer incidence, including breast cancer (*Soucise et al., 2017*), prostate cancer (*Chen et al., 2018*; *Wiggins et al., 2020*), and colorectal cancer (*Chen et al., 2018*). In the Women's Health Initiative observational study from 1994 to 2013 by *Soucise et al. (2017)* a total of 4,171 non-Hispanic whites and 235 African Americans diagnosed with breast cancer were enrolled where logistic regression was performed to investigate the connection of baseline sleep (including sleep quality) with cancer grade, hormone receptor status, cancer stage, and HER2 status. They found that AA women with "average quality sleep" (adjusted OR = 2.91; 95% CI = [1.11–7.63]) or "restless or very restless sleep" (adjusted OR = 3.74; 95% CI = [1.10–12.77]) suffered from higher breast cancer incidence when compared to those women with "sound or restful" sleep (*Soucise et al., 2017*). A study by *Wiggins et al. (2020)* collected information *via* questionnaire on the sleep quality of 5,614 men, and the results suggested that poor sleep quality was connected to high-grade prostate cancer ($OR_{T2vsT1}$: 1.39; 95% CI [1.01–1.92]) and that having trouble falling asleep at night was also connected to high-grade prostate cancer (OR: 1.51; 95% CI [1.08–2.09]) (*Wiggins et al., 2020*). Interestingly, a study by *Markt et al. (2016)* found no connection was observed between sleep quality and prostate cancer incidence. However, men who reported never feeling rested had an increased incidence of prostate cancer compared to those who reported feeling rested (*Markt et al., 2016*). Similarly, an English Longitudinal Study of Ageing cohort including 10,036 participants aged ≥50 years free of cancer from 2008, found that participants with intermediate sleep quality (hazard ratio (HR) = 1.328; 95% CI = [1.061–1.662]) or poor sleep quality (HR = 1.586; 95% CI = [1.149–2.189]) had increased incident risk of cancer compared to those with good sleep quality and that maintaining intermediate sleep quality (HR (95% CI) = 1.615 [1.208–2.160]) or poor sleep quality (HR = 1.608; (95% CI) = [1.043–2.480]) would increase the cancer incidence (*Song et al., 2020*). Thus, this prospective, nested, case-control study aimed to investigate and analyze the potential relationship between sleep quality and incidence risk of lung cancer.

## MATERIALS AND METHODS

### Study participants

The participants of this study were collected from the community cohort study "H-type Hypertension and Stroke Prevention and Control Project (HSPCP)", a controlled trial based in Lianyungang, Jiangsu Province, and Rongcheng, Shandong Province, China from 2016 to 2018. The HSPCP was an ongoing community-based, observational, non-interventional, large, prospective, and real-world registry study.

The present study mainly focused on the Lianyungang cohort. The details regarding the study design, protocol, and the HSPCP trial has been previously reported (*Hu et al., 2022*). Eligible participants were men and women between the ages of 37 and 83 who had been

diagnosed with hypertension. The diagnostic criteria at the time of screening and recruitment were: systolic blood pressure (SBP) ≥140 mmhg or diastolic blood pressure (DBP) ≥90 mmhg (three consecutive days, measured while sitting at rest and without taking antihypertensive drugs), or patients who were taking antihypertensive drugs. The trial is divided into two stages: screening, recruitment, and a 3-year follow-up period. Participants were followed up every 3 months as planned. At each follow-up, systolic blood pressure, diastolic blood pressure, medication, adverse events, and research results events were recorded. The clinical outcome initially observed in our study was the occurrence of cancer during the follow-up period. This study was approved by the Ethics Committee of the Lianyungang Precision Health Research Institute, the Institute of Biomedical Sciences of Anhui Medical University, and the Lianyungang Advanced Research Center for Cardiovascular Diseases (No. CH1059), and all participants provided written informed consent.

## Study outcomes

The lung cancer incidence information of all participants was obtained through the CDC of Lianyungang and tallied with the national medical insurance system of all inpatients' electronic links, or confirmed by active follow-up. The diagnosis of cancer was confirmed according to positive pathological results or specific clinical manifestations. Acceptable evidence of a pathology result included the original or photocopy of the pathology report, and the original or photocopy of the medical records of the hospital cited the pathology results. In the absence of pathological data, two oncologists independently reviewed this case. Cancer can only be diagnosed when two physicians make the same clinical diagnosis according to the clinical manifestations and examinations.

An independent endpoint adjudication committee reviewed and adjudicated all cancer events, and the members of this committee did not know the task of the study group.

## Study design

Using data from Lianyungang nested case-control cohort (cases = 810 and controls = 810), the selection of cases was mainly based on participants who were diagnosed with cancer during follow-up. Controls were selected from all non-cancerous participants who were alive at the time. For each case, controls were matched by sex, age, and village (±1 year) in a 1:1 ratio. After excluding 208 non-new (with a history of tumorigenesis) case-controls, 602 new cancer cases and 602 matched controls were constructed. Finally, we selected 150 new lung cancer cases and 150 matched controls (Fig. 1).

## Procedures and measures

Data collection included physical examinations and questionnaire interviews. At baseline, the measurements of height, weight, body mass index (BMI), SBP, and DBP were carried out using standardized protocols (*Hu et al., 2022*). The questionnaire was used to collect the baseline data of patients, including information about age, sex, education level, economic status, labor intensity, stress intensity, sleep patterns, alcohol consumption, smoking status, diabetes mellitus, hypertension, and cancer.

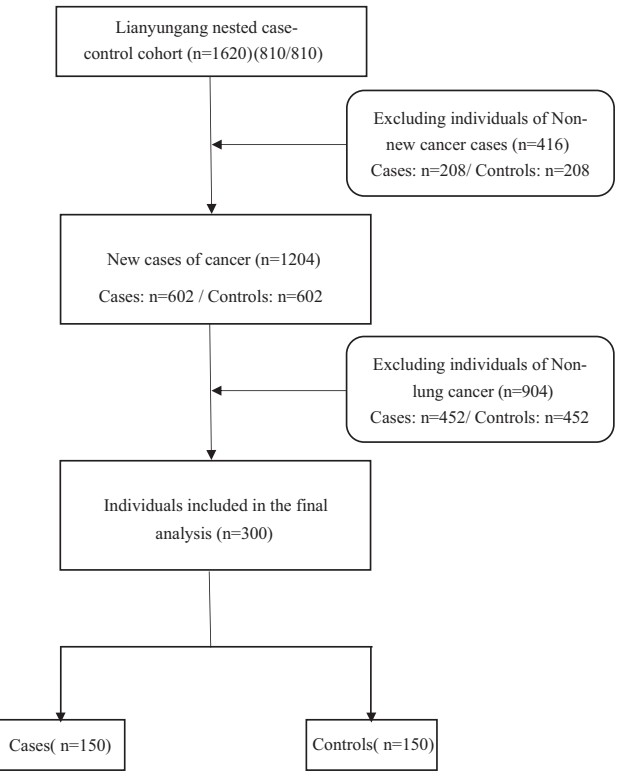

**Figure 1 Flowchart of patient selection for this study.**

## Assessment of sleep quality

To assess sleep quality, the following question was asked in the questionnaire: "How is your sleep quality at night?" Sleep quality was categorized into two groups according to the results of the questionnaire: "good", or "poor". Additionally, siesta habits, sleep patterns, and snoring were asked in the questionnaire to indirectly reflect the relationship with sleep quality. The questionnaire for these variables was as follows: "Do you usually have a siesta habit?" and the response was: "yes" or "no". "How long do you sleep on average every day?" and the answers were: "more than 8 h" or "less than 8 h". "Do you snore?" and the response was: "yes" or "no".

## Statistical analysis

Baseline continuous variables in this study are listed as mean ± standard deviation and categorical variables are listed as numbers (percentages). Differences in baseline characteristics between cases and controls were compared using generalized paired t-tests for means and chi-squared ($\chi^2$) tests for proportions.

Odds ratios (ORs) of lung cancer associated to sleep quality were estimated by conditional logistic regression models without and with adjustment for sleep duration, siesta habit, BMI, stress intensity, alcohol drinking, smoking, economic conditions, educational level, labor intensity, and snoring.

**Table 1 Characteristics of cases and controls.**

| Characteristics [a] | Controls (n = 150) | Cases (n = 150) | P value |
|---|---|---|---|
| Sex, male, n (%) | 88 (58.7) | 88 (58.7) | 1.000 |
| Age, y | 66.51 ± 9.15 | 66.51 ± 9.16 | 0.995 |
| BMI | 24.47 ± 3.28 | 24.03 ± 4.12 | 0.299 |
| Hypertension, n (%) | 85 (56.7) | 85 (56.7) | 1.000 |
| Diabetes mellitus, n (%) | 7 (4.7) | 9 (6.0) | 0.797 |
| Educational level, n (%) | | | 0.782 |
| Elementary school or lower | 233 (77.7) | 115 (76.7) | |
| Junior high school or higher | 67 (22.3) | 35 (23.3) | |
| Economic conditions, n (%) | | | 0.452 |
| High | 30 (20.0) | 24 (16.0) | |
| Moderate and low | 120 (80.0) | 126 (84.0) | |
| Labor intensity, n (%) | | | 0.433 |
| Light | 87 (58.0) | 77 (51.3) | |
| Medium | 42 (28.0) | 52 (34.7) | |
| Heavy | 21 (14.0) | 21 (14.0) | |
| Stress intensity, n (%) | 32 (21.3) | 35 (23.3) | 0.782 |
| Sleep quality, n (%) | | | 0.015 |
| Good | 82 (54.7) | 60 (40.0) | |
| Poor | 68 (45.3) | 90 (60.0) | |
| Siesta habit, n (%) | 72 (48.0) | 82 (54.7) | 0.299 |
| Sleep duration, hours | | | 0.389 |
| <8 | 97 (64.7) | 105 (70.0) | |
| ≥8 | 53 (35.3) | 45 (30.0) | |
| Snoring, n (%) | 54 (36.0) | 61 (40.7) | 0.476 |
| Smoking status, n (%) | 56 (37.3) | 62 (41.3) | 0.555 |
| Alcohol drinking, n (%) | 41 (27.3) | 29 (19.3) | 0.133 |

**Notes:**
Abbreviations: BMI, body mass index; SD, standard deviation.
[a] Data are presented as number (%) or mean ± SD.

Double-tailed $P < 0.05$ was considered to be statistically significant, except $P < 0.1$ in the interaction test. Analyses were conducted by performing the R software, version 3.6.3, and IBM SPSS Statistics, version 20.0 (SPSS Inc, Chicago, IL, USA).

# RESULTS

## Baseline characteristics

The mean age of all participants at blood collection was 66.5 ± 9.1 years; 58.7% were male. Overall, the proportion of good sleep quality in cases was significantly lower than in controls (Table 1). However, we did not observe significant differences between the two groups in sex, age, BMI, hypertension, diabetes, education level, economic status, labor intensity, stress intensity, siesta habits, sleep duration, snoring, smoking status, and alcohol consumption.

**Table 2 The relation of sleep-related factors with lung cancer incidence risk[*].**

| Lung cancer | Cases/Controls | Crude | | Adjusted | |
|---|---|---|---|---|---|
| | | OR (95% CI) | *P* value | OR (95% CI) | *P* value |
| Sleep quality [a] | | | | | |
| Good sleep quality | 60/82 | *Ref* | | *Ref* | |
| Poor sleep quality | 90/68 | 1.85 [1.146–2.98] | 0.012 | 1.83 [1.05–3.19] | 0.033 |

Note:
[*] Sleep quality.
[a] Adjusted for sleep duration, siesta habit, BMI, stress intensity, alcohol drinking, smoking, economic conditions, educational level, labour intensity, and snoring.

## Connection between sleep quality and risk of lung cancer

Within a median of 1.8 years, a total of 150 incident lung cancer cases had been observed. Sleep quality was categorized into two groups according to the results of the questionnaire, namely good-, and poor sleep quality groups. We established an adjusted clinically significant factor model, which included namely sleep duration, siesta habit, BMI, stress intensity, smoking status, alcohol drinking, economic status, educational level, labor intensity, and snoring to investigate the relationship between sleep quality and lung cancer incidence. The results showed that poor sleep quality was significantly connected to a higher lung cancer incidence (adjusted $P = 0.033$, adjusted OR = 1.83, 95% CI = [1.05–3.19]) compared to those with good sleep quality (Table 2).

## Stratified analyses

Stratified analyses were performed to identify the connection between sleep quality and lung cancer incidence in different subgroups (Fig. 2). The subgroups included age (>65, ≤65 years), sex (male, female), sleep duration (≤8 h, >8 h), siesta habit (yes, no), snoring (yes, no), stress (yes, no), smoking (yes, no), alcohol drinking (yes, no), and BMI (<24 kg/m$^2$, 24–28 kg/m$^2$, >28 kg/m$^2$). The stratified analyses only showed a significantly positive connection between poor sleep quality (*vs.* good sleep quality) and lung cancer incidence observed in smokers (*vs.* non-smokers, *P* for interaction = 0.085).

## Joint effect analysis

The joint effect analysis of sleep quality and smoking on lung cancer incidence was performed. Smokers with poor sleep quality would increase lung cancer incidence (OR = 2.79, 95% CI = [1.31–6.11]) when compared with non-smokers with good sleep quality (Table 3). However, these significant results were not observed in the good sleep quality and smoking group and the poor sleep quality and non-smoking group.

## DISCUSSION

Lung cancer is a heterogeneous disease, affected by smoking, and environmental exposure and it has a constitutional genetic or epigenetic susceptibility to the occurrence development of the disease (*Flaherty & Herridge, 2011*; *Yang et al., 2020*). In our prospective, nested, case-control study, we investigated the connection between sleep quality and lung cancer incidence. At baseline, we found that approximately 52.6% of

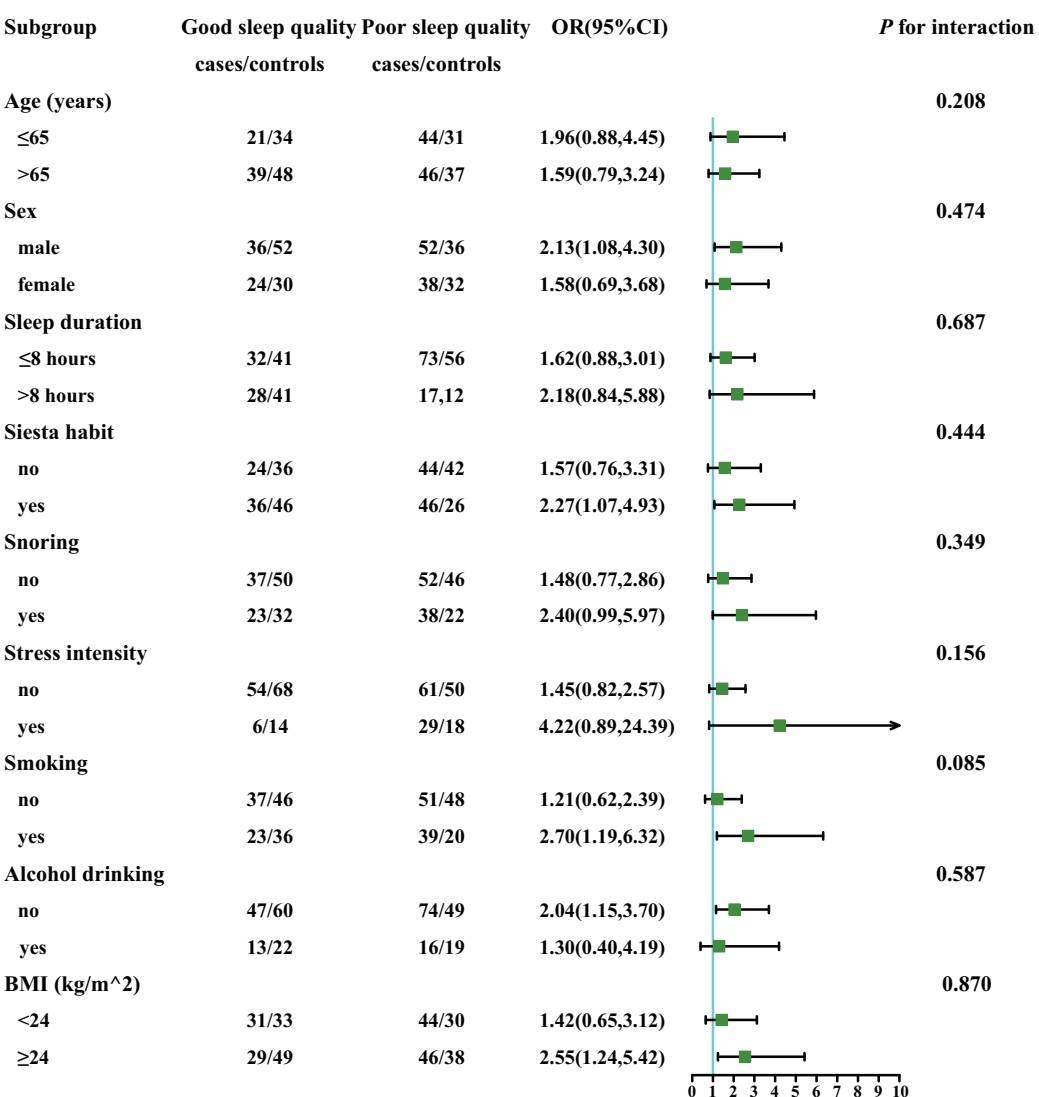

**Figure 2 The connection between sleep quality and incident risk of lung cancer in various subgroups.** Adjusted for sleep duration, siesta habit, BMI, stress intensity, alcohol drinking, smoking, economic conditions, educational level, labour intensity, and snoring.

**Table 3 The combined analysis between sleep quality and smoking on lung cancer incidence risk\*.**

| Lung cancer | Cases/ Controls | Crude | | Adjusted | |
|---|---|---|---|---|---|
| | | OR (95% CI) | P value | OR (95% CI) | P value |
| Good sleep quality & non-smoking | 37/46 | Ref | | Ref | |
| Poor sleep quality & non-smoking | 23/36 | 0.79 [0.40–1.56] | 0.506 | 0.92 [0.44–1.91] | 0.816 |
| Good sleep quality & smoking | 51/48 | 1.32 [0.74–2.38] | 0.351 | 1.26 [0.67–2.39] | 0.475 |
| Poor sleep quality & smoking | 39/20 | 2.42 [1.23–4.90] | 0.012 | 2.79 [1.31–6.11] | 0.009 |

Note:
\*Adjusted for sleep duration, siesta habit, BMI, stress intensity, alcohol drinking, smoking, economic conditions, educational level, labour intensity, and snoring.

participants self-reported that they suffered from sleep quality problems. This is higher than the results on Chinese adults in Hong Kong reported by *Wong & Fielding (2011)*, but it was higher than those found in middle-aged and elderly Chinese by *Haseli-Mashhadi et al. (2009)*. These results highlight that sleep quality has become a big problem in China.

Our multivariate logistic regression results showed that participants with poor sleep quality were met with a 1.912-fold increase the lung cancer incidence risk when compared to participants with good sleep quality. Our study also suggested that sleep quality might be a new modifiable and controllable risk factor for lung cancer. This is the first study to find that poor sleep quality is related to lung cancer occurrence, this interesting finding has not been previously reported. Although, previous studies have reported on the prognosis of patients previously diagnosed with lung cancer or patients being treated for lung cancer and its impact on sleep quality (*Chen et al., 2016*; *Lou et al., 2017*; *Papadopoulos et al., 2019*).

Sleep quality is an important cancer-related incident risk factor. Many reports have found that sleep quality is also connected to the incidence of other cancer types, including breast cancer (*Soucise et al., 2017*) and prostate cancer (*Sigurdardottir et al., 2013*; *Wiggins et al., 2020*). Our suggestive findings combined with these findings help support the positive relationship between poor sleep quality and an increasing lung cancer incidence.

Poor sleep quality might cause the incident risk of cancer by inducing sleep disruption, melatonin, and lifestyle disturbance (*Fritschi et al., 2011*). Melatonin takes part in the anti-inflammatory and immune-modulatory effects (*Soucise et al., 2017*). Increasing evidence supports the use of melatonin as a drug against the initiation, progression, and metastasis of cancer, and it plays a major part in interfering with various cancer characteristics, including angiogenesis, continuous proliferation, metastasis, and resistance to cell death (*Talib, 2018*). In addition, sleep disruption and circadian disruption might lead to oncogenesis by influencing the immune system, inducing DNA damage, and disrupting the metabolism (*Samuelsson et al., 2018*).

In our study, the results of the subgroup analysis showed that smoking and sleep quality are strongly positively related to each other, and the combined effect analysis suggests that smokers with poor sleep quality suffered from a 2.79-fold cancer risk when compared with non-smokers with good sleep quality. Smoking is currently recognized as a risk factor for lung cancer. Smoking can cause lung cancer. This process is driven by more than 60 carcinogens in cigarette smoke. These carcinogens directly damage and change DNA. In addition, smoking will increase the burden of gene mutations, the heterogeneity of cells, and driver gene mutations (*Yoshida et al., 2020*). Sleep is a reversible state that maintains the physical and psychological processes of the human body. It is shown that smoking can increase the incident risk of poor sleep quality (*Kabrita, Hajjar-Muca & Duffy, 2014*; *Wu et al., 2018*). A meta-analysis of 24 prospective cohort studies of prostate cancer, from 21,579 cancer participants suggested that smoking was connected to prostate cancer incidence risk (*Huncharek et al., 2010*). The same results also were observed by *Rohrmann et al. (2007)* and *Giovannucci et al. (1999)*. Complaints related to sleep quality, including increased sleep latency and decreased total sleep time, are common among smokers (*Cohrs et al., 2014*; *Zhang et al., 2006*). Smoking can lead to difficulty falling asleep and sleep

interruption, which is probably due to the excitatory effect of nicotine (*Wetter & Young, 1994*). In addition, more severe nicotine dependence is associated with sleep disorders (*Branstetter et al., 2016*; *Cohrs et al., 2014*; *Hamidovic & de Wit, 2009*). In a laboratory study, people smoked more after a period of sleep deprivation than after normal sleep (*Patel et al., 2010*). It is assumed that sleep deprivation may increase the intrinsic value of cigarettes because smoking is expected to combat fatigue (*Hoggard & Hill, 2018*). Among Japanese civil servants, the overall sleep quality of men who have quit smoking is significantly worse than that of non-smokers, and the overall sleep quality of female smokers is also significantly worse than that of non-smokers (*Takamatsu et al., 2010*). So, it is necessary for us to associate smoking with the quality of sleep, because they are closely related. But there were no related reports investigated the interaction between the sleep quality and smoking in lung cancer incident risk. These two factors influence each other. This result is of great guiding significance and reduce the lung cancer incidence.

The strengths of our study include high-quality data, and a prospective, nested, case-control study design. Major covariates were available, which allowed us to investigate independent links between sleep quality and lung cancer incidence. However, limitations also existed and need to be mentioned. Firstly, the related information on sleep quality was self-reported. Participants' understanding and feelings might have contributed to some bias in the results. In the future, sleep quality needs to be further quantified or diagnosed by professional doctors. Secondly, the study population sample was small, thus a larger sample is necessary to validate this finding in future studies. Of course, we may also need to include different ethnic, provincial, and other relevant factors for analysis. Finally, this study is only a cross-sectional study, and a longitudinal dynamic observation is still needed.

## CONCLUSION

Using a nested case-control design, the first time, we found that poor sleep quality was positively connected to lung cancer incidence. In addition, for those with poor sleep quality, smoking increased lung cancer incidence.

### Funding

This work was supported by the National Key Research and Development Program (grant numbers [2017YFC1309200] and [2022YFC2009600]) and the Beijing Municipal Science and Technology Commission [grant number SCW2018-06]. The funders had no role in study design, data collection and analysis, decision to publish, or preparation of the manuscript.

### Grant Disclosures

The following grant information was disclosed by the authors:
National Key Research and Development Program: 2017YFC1309200 and

2022YFC2009600.
Beijing Municipal Science and Technology Commission: SCW2018-06.

## Competing Interests

Yun Song is employed by Shenzhen AUSA Pharmed Co Ltd. All authors declare that they have no competing interests.

## Author Contributions

- Guo-Tian Ruan conceived and designed the experiments, performed the experiments, analyzed the data, prepared figures and/or tables, authored or reviewed drafts of the article, and approved the final draft.
- Ya-Ping Wei conceived and designed the experiments, analyzed the data, prepared figures and/or tables, authored or reviewed drafts of the article, and approved the final draft.
- Yi-Zhong Ge analyzed the data, prepared figures and/or tables, authored or reviewed drafts of the article, and approved the final draft.
- Li-Shun Liu analyzed the data, authored or reviewed drafts of the article, and approved the final draft.
- Zi-Yi Zhou analyzed the data, authored or reviewed drafts of the article, and approved the final draft.
- Sultan Mehmood Siddiqi analyzed the data, authored or reviewed drafts of the article, and approved the final draft.
- Qiang-Qiang He analyzed the data, authored or reviewed drafts of the article, and approved the final draft.
- Shu-Qun Li analyzed the data, authored or reviewed drafts of the article, and approved the final draft.
- Jia-Feng Xu analyzed the data, authored or reviewed drafts of the article, and approved the final draft.
- Yun Song analyzed the data, authored or reviewed drafts of the article, and approved the final draft.
- Qi Zhang analyzed the data, authored or reviewed drafts of the article, and approved the final draft.
- Xi Zhang analyzed the data, authored or reviewed drafts of the article, and approved the final draft.
- Ming Yang analyzed the data, authored or reviewed drafts of the article, and approved the final draft.
- Ping Chen analyzed the data, authored or reviewed drafts of the article, and approved the final draft.
- Yong Sun analyzed the data, authored or reviewed drafts of the article, and approved the final draft.
- Xiao-Bin Wang conceived and designed the experiments, analyzed the data, authored or reviewed drafts of the article, and approved the final draft.

- Han-Ping Shi analyzed the data, authored or reviewed drafts of the article, and approved the final draft.
- Bin-Yan Wang analyzed the data, authored or reviewed drafts of the article, and approved the final draft.

## Data Availability

The raw data are available in the Supplemental Files.

## Supplemental Information

Supplemental information for this article can be found online at http://dx.doi.org/10.7717/peerj.16540#supplemental-information.

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
