# Peer review of "Poor sleep quality association with higher lung cancer risk: a nested case-control study"

_PeerJ, doi:10.7717/peerj.16540_

## Round 0.1 · original submission · Major Revisions

Please see the comments from the two reviewers and revise the manuscript accordingly. As highlighted by Reviewer #1, please consider using professional proofreading.

·

Basic reporting

A well-written article with minor grammatical errors. Please consider using professional proofreading services or software ( for example, Grammarly).

Experimental design

no comment

Validity of the findings

no comment

Additional comments

This well-organized and informative research article investigates a novel area of study: the possible relationship between sleep quality and lung cancer incidence risk. The study design of the nested case-control study was well described. The justification for the 1:1 ratio was clearly understood.

Some of these warrants clarifications:
1. Are there any other causes of sleep abnormalities other than insomnia (as mentioned in the introduction) that might be present as confounding variables?

2. It might be worth noting why a p-value of less than 0.10 was chosen as the threshold for statistical significance, as a p-value of less than 0.05 is typically used in most studies.

3. Validity of the sleep questionnaire

Reviewer 2 ·

Basic reporting

The authors conducted a study to understand the correlation between poor sleep quality and lung cancer on a small subset of population from the Lianyungang province of China.
- On line 64-65, please mention what some of the important risk factors contributing to increased chances of lung cancer are
- Please mention that years for European and East Asia. If the comparison is from 2002-2012, author must specify that at the start of the statement on line 73
- Line 81-82 needs to be rewritten to sound more cohesive. Consider formatting to “Poor sleep quality decreases floating melatonin in the system”

Experimental design

The authors collected data from a pre-studied cohort of individuals from three provinces in China: Lianyungang, Jiangsu province, and Rongcheng, Shandong province. However, for the scope of this study, the population was limited to samples from Lianyungang province.
- Were the individuals surveyed about their health after the study concluded?
- The authors should provide clarification on lines 131-133. What does "non-new cases controls" refer to? Were there 208 individuals who did not develop cancer during the follow-up period? Please state this clearly. What is meant by "208/208"?
- The authors need to clearly explain the sampling process. Based on the information provided, it appears that out of the 810 participants, 208 patients did not develop cancer while 602 did. Were these 208 patients used as controls? If so, where were the 602 matched controls obtained from?
- Furthermore, it is unclear where the 150 samples came from.
- Considering that smoking is the leading cause of lung cancer development, were there considerations for dividing the experimental group into smokers and non-smokers? It is important to clearly mention the different subgroups in the study under the Experimental Design section.
Clarity on these aspects of the experimental design will enhance the understanding of the study methodology and strengthen the validity of the findings.

Validity of the findings

- Line 166: Could the notation (9.1) refer to the standard deviation?
- The introductory comparison of other studies with similar conclusions should be moved to the Introduction section rather than the Discussion section. The Discussion section should focus on analyzing the findings of the current study.
- While the authors performed robust statistical analyses to support their hypothesis, additional data is needed to strengthen the hypothesis. Considering the strong claim made that poor sleep quality is correlated with lung cancer, it is crucial to discuss the possibility of an inverse correlation, particularly within the smoking population.
- Has any significant analysis been conducted to establish that smoking was not the sole causal factor in cancer development? The study does not provide sufficient evidence to rule out other interacting factors contributing to the development of lung cancer.
- Lines 257-258 suggest a 2.79-fold higher risk of cancer in smokers with poor sleep quality compared to non-smokers with good sleep. However, there is no evidence indicating that poor sleep quality was not a consequence of smoking.
- Moreover, the study's sample size is limited to a small subset of the population. To enhance the study's generalizability, I recommend expanding the sample to include diverse ethnicities, provinces, and other relevant factors.
- Additionally, the study solely relied on qualitative questionnaires provided to the patients. The qualitative nature of the surveys and questionnaires may not always provide a comprehensive representation of the true results. To strengthen the hypothesis, authors might want to consider incorporating other quantitative methods to track the sleep patterns and heart rate of patients. This could provide more objective and reliable data, enhancing the validity of the findings.

Additional comments

- In line 61, the author may consider rephrasing the sentence to: "One of the most significant reasons for lower life expectancy is cancer."
- Authors are advised to carefully review the paper for grammatical errors to enhance its cohesiveness and improve readability for an international audience.
- When citing multiple sources, adhere to proper journal guidelines. For example, instead of citing (10-12), the author should cite [10][11][12].
- Was it 52.7% of the 58.7% male participants who reported experiencing sleep problems, or was it 52.7% of the entire population? (Line 46)
- Did we compare smokers with good sleep quality? (Lines 51-52)
- Line 55: Did we investigate whether the correlation held true the other way around? Did smoking have any influence on sleep quality?

Annotated reviews are not available for download in order to protect the identity of reviewers who chose to remain anonymous.

---

## Round 0.2 · accepted · Accept

Thank you for addressing all the reviewers' comments and for revising the paper.
I am happy to accept this paper for publication. Congratulations!

·

Basic reporting

No comment

Experimental design

No comment

Validity of the findings

No comment

Reviewer 2 ·

Basic reporting

- Clear, easy to understand article
- Authors have edited the article grammatically

Experimental design

- Experimental design is much more defined on the revision
- Authors clarified the controls and sampling process clearly
- it is easier to follow the methodology

Validity of the findings

- Author provided more data to support claim for smoking population
- Author clarified the involvement of other factors that can cause lung cancer
- Author has added clarification on lack of diversity in sample set

In general, this paper offers an adequate blend of qualitative and quantitative evidence to establish a correlation between diminished sleep quality and an elevated risk of lung cancer, albeit with some important reservations. I believe that this research has the potential to make a valuable contribution to the respective field.